# Phylogeography and Population Variation in *Prunus discoidea* (*Prunus* subg. *Cerasus*) in China

**DOI:** 10.3390/plants13172535

**Published:** 2024-09-09

**Authors:** Xiangzhen Chen, Shucheng Gao, Hong Yang, Wenyi Fu, Siyu Qian, Xianrong Wang, Xiangui Yi

**Affiliations:** 1Co-Innovation Center for Sustainable Forestry in Southern China, College of Life Sciences, Nanjing Forestry University, Nanjing 210037, China; chenxiangzhen0508@163.com (X.C.); 18762143959@163.com (S.G.); yanghong991007@gmail.com (H.Y.); fufu0125@163.com (W.F.); qiansy1011@163.com (S.Q.); wangxianrong66@njfu.edu.cn (X.W.); 2Cerasus Research Center, College of Life Sciences, Nanjing Forestry University, Nanjing 210037, China

**Keywords:** *Prunus discoidea*, population genetic diversity, population genetic structure, phylogeography

## Abstract

*Prunus discoidea* is a unique cherry blossom germplasm resource native to China. It is widely distributed across the provinces of Anhui, Zhejiang, Jiangxi, Jiangsu, and Henan, with significant variation. We employed phylogeographic analysis to reveal the evolutionary history of *P. discoidea* to better understand its genetic diversity and structure. This study provides more accurate molecular insights for the effective conservation and utilization of this germplasm resource. We conducted a phylogeographic analysis of 348 individual plants from 13 natural populations using three fragments (*rpoB*, *rps16*, and *trnD–E*) of chloroplast DNA (cpDNA) and one fragment (ITS) of ribosomal DNA. The results revealed that *P. discoidea* demonstrates a significant level of genetic diversity (*H_d_* = 0.782; *R_d_* = 0.478). Gene flow among populations was limited, and the variation within populations was the main source of genetic diversity in *P. discoidea* (among populations: 34.26%, within populations: 65.74%). Regarding genetic differences among populations, *N_st_* (0.401) showed greater differences than *G_st_* (0.308; *p* < 0.05), demonstrating that there was a significant geographical structure of lineage. One lineage was the central region of Anhui and the western region of Hubei. The other lineage was the Jiangsu region and the Zhejiang region. *P. discoidea* diverged from *Prunus campanulata* approximately 1.5 million years ago, during the Pleistocene epoch. This study provides a scientific theoretical basis for the conservation and utilization of germplasm resources of *P. discoidea*.

## 1. Introduction

*Prunus discoidea* is a member of the *Prunus* subg. *Cerasus* in the Rosaceae family. It is an excellent germplasm resource endemic to China [1]. The branches of *P. discoidea* are graceful and spreading, with pink flowers that bloom early in the season. It is widely distributed in Anhui, Zhejiang, Jiangxi, Jiangsu, and Henan provinces [2], growing in valley forests or streamside thickets at elevations of 200–1100 m [3]. It has a wide distribution and significant variation. Currently, the research on *P. discoidea* mainly focuses on the resources, community structure, species diversity, and morphological characteristics. For example, Nan et al. showed that the ecological niches of the *P. discoidea* communities in four areas (Huang Mountain, Baiyun Mountain, Tianmu Mountain, and Lu Mountain) had a high degree of overlap [4]. Yan et al. indicated that *P. discoidea* primarily inhabited the central and eastern regions of China [3]. Fu et al. indicated that *P. discoidea*, *Prunus cantabrigiensis*, and *Prunus helenae* were grouped into one clade [5]. Shang et al. indicated that there was a high level of genetic differentiation among *P. discoidea* populations, with gene flow being obstructed. Additionally, four populations were divided into two clades: one consisting of Huang Mountain and Lu Mountain, and the other comprising Tianmu Mountain and Baiyun Mountain [6]. However, research on the phylogeography of *P. discoidea* is still lacking, and its biogeography and trait evolution are not yet clear.

Phylogeography is often referred to as molecular phylogeography. It is a field of study concerned with the principles and processes governing the geographic distributions of genealogical lineages, especially those within and among closely related species [7]. The concept was first introduced by Avise in 1987 [8]. It enables a more precise depiction of genealogical geographic structures, variances in geographic distributions, and the spatial and temporal dynamics underlying speciation, utilizing advanced bioinformatic tools [9]. As DNA sequence data continue to expand and resources become increasingly abundant, research in plant phylogeography has progressed from initially examining changes in gene frequencies to using microsatellite marker technology and single-nucleotide polymorphism data to cluster populations. Subsequently, phylogeographic research has employed a combination of microsatellite genetic markers or a small number of cpDNA and mtDNA sequences [10]. The study of the phylogeography of *Prunus* subg. *Cerasus* in China has only recently commenced, constrained by sampling time, costs, and molecular marker technologies. Currently, a combination of chloroplast fragments and nuclear gene fragments is often used to leverage their respective advantages for comprehensive analysis [11]. In *Prunus* subg. *Cerasus*, chloroplast sequence fragments are mainly represented by *matK* [12] fragments and non-coding intergenic spacer regions such as *trnD–trnE* [13] and *trnL–trnF* [14]. Within the nuclear genome, the ITS sequences are considered core markers for identifying plants within the *Prunus* subg. *Cerasus*.

In this study, based on a comprehensive survey of wild populations of *P. discoidea* and systematic sampling, we conducted a phylogeographic analysis of *P. discoidea*. Chloroplast DNA sequences and nuclear ribosomal internal transcribed spacer (ITS) sequences were used to analyze the genetic diversity and genetic structure of *P. discoidea* and an integrative method employed to trace its evolutionary history. These findings provide a theoretical foundation for future strategies related to the conservation and utilization of *P. discoidea* resources.

## 2. Results

### 2.1. Population Genetic Diversity

Three cpDNA fragments, namely, *rps16, rpoB*, and *trnD–E*, had a combined length of 1886 bp, with fragment lengths of 887 bp, 440 bp, and 632 bp, respectively. Based on the concatenated sequences, nine variable sites were detected. Three mutation sites were detected for each of *trnD–E*, *rpoB*, and *rps16*. Seventeen chloroplast haplotypes (H1–H17) were recovered from the 13 populations (Table 1). At the species level, the overall population haplotype diversity (*H_d_*) was 0.782, and the nucleotide diversity (*P_i_*) was 0.00104. At the population level, the haplotype diversity (*H_d_*) ranged from 0.000 to 0.891, and the nucleotide diversity (*P_i_*) ranged from 0.00000 to 0.00113. At the regional level, the haplotype diversity of the two geographical regions ranged from 0.703 to 0.807, and the nucleotide diversity ranged from 0.00087 to 0.0013 (Table 2).

The sequence length of the nrDNA fragment measured was 692 bp, and 13 variable sites were detected. Six ribotypes (R1–R6) were recovered from the 13 populations (Table 3). At the species level, the overall ribosomal diversity (*R_d_*) of *P. discoidea* was 0.478, and the nucleotide diversity (*P_i_*) was 0.00451. At the population level, the ribosomal diversity (*R_d_*) ranged from 0.000 to 0.756, and the nucleotide diversity (*P_i_*) ranged from 0.00000 to 0.00571. At the regional level, the ribosomal diversity of the two geographical regions ranged from 0.348 to 0.530, and the nucleotide diversity ranged from 0.00336 to 0.00489 (Table 4).

### 2.2. Population Genetic Structure

The population differentiation index (*F_st_*) of *P. discoidea* at the cpDNA level was 0.34264, signifying a significant level of differentiation. Among populations, genetic variation accounted for 36.27%, while within populations it was 63.73%. Genetic variation within populations slightly exceeded the variation among populations, although the values were similar. As shown by the AMOVA results, genetic variation among regional groupings was 12.47%, genetic variation among populations within regional groupings was 25.57%, and genetic variation within populations in regional groupings was 61.96%. The genetic variation within populations among regional groupings was greater than the genetic variation among populations. The genetic differentiation parameters of *P. discoidea* (*N_st_* = 0.40104, *G_st_* = 0.30809, *p* < 0.05) indicated population substructure. Genetic differentiation was detected in both geographic regions: eastern China (*N_st_* = 0.34538, *G_st_* = 0.22369, *p* < 0.05) and central China (*N_st_* = 0.3413, *G_st_* = 0.32690, *p* < 0.05). The phylogeographic structure was detected in both locations (Table 5).

The population differentiation index (*F_st_*) of *P. discoidea* at the ITS level was 0.57621, which suggested a high degree of species differentiation in *P. discoidea*. The genetic variation among populations accounted for 51.26%, while the genetic variation within populations was 48.74%, meaning that the genetic variation among populations was slightly higher than the genetic variation within populations. Genetic variation among regional groupings was 4.57%, the genetic variation among populations within regional groupings was 25.57%, and genetic variation within populations in regional groupings was 61.96%, indicating that the genetic variation among populations within each geographic group was slightly higher than the genetic variation within populations (Table 5).

### 2.3. Phylogeographic Structure

ZJS (Xianning, Hubei) and YZH (Lu’an, Anhui) had the most haplotypes, with eight haplotypes each, while BYS (Lishui, Zhejiang) and LS (Jiujiang, Jiangxi) had the fewest, with only one haplotype each. Regarding the frequency of haplotypes, H2 had the highest distribution frequency, totaling 133 individuals, and the lowest frequency was observed for H16 and H17, each with a distribution of three individuals (Figure 1A). Based on the TCS haplotype network diagram, H2 was located in the central position of the network diagram, with a wide distribution range and a higher proportion of individuals in the population. This suggested that it was an ancient haplotype. Haplotypes H3 and H4 were located in sub-core positions and had a broader distribution, and hence they were classified as sub-ancient haplotypes (Figure 1B). *Prunus padus*, *Prunus salicina*, *Prunus mume*, *Prunus mahaleb*, and *Prunus cerasoides* were used as outgroups. The haplotype phylogenetic tree of *P. discoidea* based on the maximum likelihood method and Bayesian method showed a consistent topological structure. The 17 haplotypes formed a highly supported monophyletic group. The phylogenetic tree diverged into two distinct lineages, which were consistent with the clustering results of the haplotype TCS network. H1 and H12–H15 diverged into one clade, corresponding to the eastern lineage, while H6–H11 and H16–H17 diverged into another clade, corresponding to the central lineage (Figure 2A,B).

YZH had the most ribosomal types, with four, ZJG (Jingmen, Hubei) and DMS (Lin’an, Zhejiang) each had three ribotypes, and ZJS, LS, YTS (Lianyungang, Jiangsu), LKY (Nanyang, Henan), and THC (Huanggang, Hubei) had the fewest, with only one ribotype each (Figure 1C). R1 was located in the center of the network diagram, contained the most individuals, and was found in all populations except for Yuntai Mountain, suggesting that it was an ancient haplotype. The remaining ribotypes had all further mutated, establishing interconnections among different regions (Figure 1D). *P. padus*, *P. salicina*, and *Prunus pseudocerasus* were used as outgroups. The haplotype phylogenetic tree based on the maximum likelihood method and Bayesian method showed a consistent topological structure. The six ribotypes identified formed two distinct groups: one east lineage and one central lineage. R2 and R3 were ribotypes unique to the eastern region, and R4–R6 were ribotypes unique to the central region (Figure 3A,B).

### 2.4. Molecular Dating and Historical Dynamics

The *rps*16 fragments of 18 species from the Rosaceae family, including *Dryas octopetala*, *Sanguisorba filiformis*, *P. mahaleb*, *Prunus campanulata*, *P. pseudocerasus*, *Prunus maximowiczii* and *Prunus dielsiana*, were downloaded from NCBI. Using PhyloSuite version 1.2.3, the sequences were aligned with Hap2 of *P. discoidea*. Using four molecular fossils (Rosaceae crown = 101.6 mya [15], Rosoideae = 75.78 mya [16], Maleae = 50.06 mya [16], *Prunus* + *Cerasus* = 28.21 mya [17]; 95% HPD) to calibrate the timeline, we constructed a phylogenetic tree at the family level. Finally, it was estimated that *P. discoidea* diverged from *P. campanulata* approximately 1.5 million years ago (mya) during the Pleistocene epoch (95% HPD; 0–6.27 mya) (Figure 4).

The historical dynamics of *P. discoidea* were analyzed mainly by neutral tests and mismatch analysis. Tajima’s *D* and Fu’s *F_s_* neutrality tests were conducted at the species level and in each geographical group. Based on the results of cpDNA molecular markers, Tajima’s *D* value for the ZJG population was negative and Fu’s *F_s_* value was positive, with *p* > 0.05, indicating non-significance. Tajima’s *D* values for ZJS and YZH were positive, while Fu’s *F_s_* values were negative, with *p* > 0.05, indicating non-significance. Tajima’s *D* and Fu’s *F_s_* values for LS and BYS were both zero. Tajima’s *D* and Fu’s *F_s_* values of the other eight populations were all positive, with *p* > 0.05, indicating non-significance (Appendix A). The results of population tests showed that Tajima’s *D* value was positive and Fu’s *F_s_* value was negative (*p* > 0.05). In both the eastern and central geographical groups, Tajima’s *D* values were positive, Fu’s *F_s_* values were negative, and *p* > 0.05, indicating non-significance (Appendix A). In conclusion, *P. discoidea* did not experience population expansion events or bottleneck effects. However, the mismatch analysis of *P. discoidea* populations showed a unimodal curve (Appendix A), with an SSD value of 0.01938 and Hrag value of 0.05561, with *p* > 0.05, consistent with the hypothesis of population expansion (Appendix A). This indicated that *P. discoidea* populations experienced a recent population expansion event.

Based on the results of ITS molecular markers, Tajima’s *D* and Fu’s *F_s_* values for the seven populations of BYS, DMS, HS (Huangshan, Anhui), SMS (Ningbo, Zhejiang), ZJG, LCS (Yixing, Jiangsu), and YZH were all positive, with *p* > 0.05, indicating non-significance. Tajima’s *D* value and Fu’s *F_s_* value in the BMQ (Jiande, Zhejiang) population were negative, with *p* > 0.05, which was not significant. Tajima’s *D* and Fu’s *F_s_* values for the five populations of ZJS, LS, YTS, LKY, and THC were zero, with *p* > 0.05, indicating non-significance (Appendix A). Additionally, Tajima’s *D* and Fu’s *F_s_* values of the population and the eastern and central geographical groups were positive, with *p* > 0.05 (Appendix A). The mismatch distribution curve of *P. discoidea* was multimodal, with *p* > 0.05, indicating non-significance. The mismatch analyses of both the populations and the geographical groups showed bimodal curves (Appendix A). In summary, these results suggested that the populations of *P. discoidea* did not experience rapid expansion or contraction events recently.

## 3. Discussion

### 3.1. Genetic Diversity and Population Genetic Structure

Genetic diversity is defined as the variety of genetic materials and genetic information within all biological individuals. This includes gene mutations among different populations of the same species as well as genetic differences within the same population. This is of great importance for the maintenance and propagation of species, adaptation to the environment, and resistance to adverse environmental conditions and disasters. Liu et al. used nSSR and cpDNA to analyze the genetic diversity of Chengbutong tea and showed that the genetic diversity (*H_d_*) was 0.732 [18]. Li et al. used cpDNA non-coding sequencing to study the genetic diversity of *Salix psammophila* and showed that the genetic diversity (*H_d_*) was 0.737 [19]. Li et al. studied the phylogenetic relationship of the genus *Disanthus* distributed disjunctively in China and Japan based on cpDNA sequences and showed that the genetic diversity (*H_d_*) among populations was 0.725 [20]. The genetic diversity (*H_d_*) within the population of *P. discoidea* was 0.782, the variation in haplotype diversity (*H_d_*) among the different populations ranged from 0.000 to 0.891, and the variation in nucleotide diversity (*P_i_*) ranged from 0.00000 to 0.00114. These results showed that *P. discoidea* populations had a relatively high level of genetic diversity. The research results were similar to those of previous studies on *Prunus serrulate* [21], *P. dielsiana* [22], and *Prunus conradinae* [23]. The ITS genetic diversity (*R_d_*) within *P. discoidea* was 0.478, and the nucleotide diversity (*P_i_*) was 0.00451. The variation in ribosomal diversity (*R_d_*) ranged from 0.175 to 0.756, and the variation in nucleotide diversity (*P_i_*) ranged from 0.00051 to 0.00571. The findings showed that the genetic diversity was lower compared to *Prunus tomentosa* [24], *P. pseudocerasus* [25], and *Prunus avium* [26]. However, both markers indicated that *P. discoidea* exhibited a high level of genetic diversity, which is presumed to be related to the growth environment and the distribution of its habitats. *P. discoidea* was distributed in central and eastern China, located on the third step of China’s geographical terrain, and was concentrated in the middle and lower reaches of the Yangtze River. The region is characterized by flat terrain with no mountainous barriers. At the same time, the warm and humid climate conditions maintained their genetic diversity, and the activities of birds and humans made hybridization and self-pollination within or among neighboring populations possible.

As shown in Table 5, based on three cpDNA fragments, the genetic variation among populations was lower than that within populations. Based on ITS fragments, the genetic variation among populations was higher than that within populations. The genetic differentiation coefficients for both molecular markers reached significant levels, and gene flow among populations of *P. discoidea* was relatively weak. Therefore, this study suggests that the variation in the population of *P. discoidea* mainly arose from the variation within populations. The study by Shang et al. on the analysis of population diversity in *P. discoidea* using SSR markers was consistent with the findings presented here [6]. Based on three cpDNA fragments, the results for *P. discoidea* populations and two geographic groups separately showed that the genetic differentiation coefficients reached significant levels. This finding was consistent with the genetic differentiation parameters of nrDNA markers, indicating the presence of phylogeographic structure within the geographic groups and populations of *P. discoidea*. The research results were similar to those of previous studies on *P. serrulate* [21], *P. dielsiana* [22], and *P. conradinae* [23]. This result was consistent with the habits of *P. discoidea*, which in its natural state tended to individual scattered distribution. *P. discoidea* was typically distributed on cliffs and river valleys, with long-distance seed dispersal primarily relying on bird and human activities.

### 3.2. Geographical Structure

Based on the geographical distribution of haplotypes and ribotypes, there were at least two genetic lineages within *P. discoidea*. One lineage included Anhui and areas to the west of Hubei. This division was based on the unique haplotypes H6–H11 and H16–H17, which were primarily distributed in THC, YZH, and ZJS. The other lineage included the regions of Jiangsu and Zhejiang, based on the unique haplotypes H1, and H12–H13, which were primarily distributed in BMQ, HS, YTS, and SMS. The reason for this might have been that *P. discoidea* was mainly distributed in central and southeast China. Furthermore, the sampling points were situated in the third step of the Chinese geographical map, with flat terrain and the presence of only two natural barriers, namely, Huang Mountain and Lu Mountain. This formed two distinct lineages in the eastern and central regions. The research results were consistent with those of previous studies on *P. serrulata* [21].

### 3.3. Historical Dynamics of P. discoidea Group

Based on neutrality tests for *P. discoidea* populations and geographic groups, the results indicated that neither of the two molecular markers pointed to population expansion or contraction events. However, the mismatch distribution analysis based on cpDNA molecular markers for *P. discoidea* population and geographic groups showed a unimodal curve. Both the SSD value of 0.01938 (*p* = 0.18000 > 0.05) and the Hrag value of 0.05561 (*p* = 0.33000 > 0.05) are consistent with the hypothesis of the population expansion model. This suggested that *P. discoidea* had experienced a population expansion event. The results contradicted those of the neutrality tests. However, according to Appendix A, the population size of *P. discoidea* was *θ*_0_ = 0.00176 before the outbreak and *θ*_1_ = 5.33203 after the outbreak. The change in effective population size (*θ*_0_ − *θ*_1_ = 5.33203 − 0.00176) was large, so it was considered that *P. discoidea* had recently experienced a population expansion event. This result was consistent with a phylogeographic study of *Xanthopappus subacaulis* in the northeastern Qinghai–Tibet Plateau conducted by Zhang Yang et al. [27].

Based on the phylogenetic tree, *P. discoidea* diverged from *P. campanulata* approximately 1.5 million years ago, during the Pleistocene epoch. In phylogeographic studies, glacial refugia were usually detected through the high diversity of haplotypes and major lineages within species populations [28,29]. We inferred that *P. discoidea* had two mainly glacial refugia, namely, Dabie Mountain around Yanzihe Canyon in Anhui and Qingliang Peak in Zhejiang. Following the glacial period, *P. discoidea* spread from these refuges, with its diffusion route roughly spanning Anhui–Henan–Hubei–Jiangxi or directly from Anhui to Jiangxi. Another diffusion route was from Zhejiang–Anhui–Jiangsu or directly from Zhejiang to Jiangsu, resulting in the formation of two lineages in the middle and east, and the current distribution pattern of *P. discoidea*. The findings of this research were largely consistent with those of previous phylogeographic studies of *P. dielsiana* [30] and *P. serrulate* [21].

## 4. Materials and Methods

### 4.1. Plant Materials

The distribution data of *P. discoidea* is primarily based on the Chinese Virtual Herbarium (CVH: https://www.cvh.ac.cn/ (accessed on 10 October 2020)) and published academic papers. For distribution points with accurate specimen records, but lacking latitude and longitude data, LocaSpaceViewer (http://www.locaspace.cn/ (accessed on 20 November 2020)) was used to ascertain the coordinates, thereby enhancing the precision of the specimen information. DIVA-GIS was used to filter the obtained data, deleting duplicate records and those with collection points that were too close to each other. From 2020 to 2022, a total of 348 samples from 13 populations of *P. discoidea* were collected through two consecutive years of field investigation and sample collection (Table 6 and Table 7, Figure 5). Within each population, 10 to 35 individuals were randomly selected, each at least 30 m apart. For each individual, 5 to 10 mature, healthy, and intact small leaves were collected. Then, the samples were rapidly placed in silica gel for drying. Finally, the samples were put into the refrigerator at −20 °C for use.

### 4.2. DNA Extraction, Polymerase Chain Reaction (PCR) Amplification, Sequencing, and Sequence Alignment

According to previous research, leaves from the *Prunus* subg. *Cerasus* are known to contain significant amounts of polysaccharides and polyphenols, making DNA extraction quite challenging. Therefore, the DNA of *P. discoidea* was co-extracted using a polysaccharide polyphenol reagent kit (Tiangen Biotechnology Co., Ltd., Shanghai, China) [21] and a modified CTAB method [31]. The concentration and purity of the extracted DNA were assessed via 1% agarose gel electrophoresis. Samples that did not meet the quality criteria were excluded, and the qualified DNA samples were stored at −80 °C for preservation. These samples were shipped to Shenggong Bioengineering (Shanghai) Co., Ltd. for sequencing, and the haplotypes were obtained for lineage geographical analysis. By reviewing the relevant literature and accessing the NCBI website (https://www.ncbi.nlm.nih.gov/ (accessed on 20 November 2022)), universal primers for different sequences of cpDNA and nrDNA from the *Prunus* subg. *Cerasus* was selected and collected. Three pairs of cpDNA universal primers are respectively *rps16* (F: GTGGTAGAAAGCAACGTGCGACTT; R: TCGGGATCGAACATCAATTGCAAC) [13], *rpoB* (F: AAGTGCATTGTTGGAACTGG; R: CCCAGCATCACAATTCC) [32], *trnD–E* (F: ACCAATTGAACTACAATCCC; R: AGGACATCTTCAAGGAG) [33], and a pair of nrDNA sequence fragment ITS (F: TCCTCCGCTTATTGATATGC; R: GGAAGGAGAAGTCGTAACAAGG) [34], to be used for determining the genetic diversity and population structure of *P. discoidea*. A 25 μL PCR amplification reaction system was constructed using 1.0 μL of DNA template, 1.0 μL (10 μmol L^−1^) each of upstream and downstream primers, 12.5 μL of 2 × PCR Master Mix, and 9.5 μL of ddH_2_O [35]. The PCR amplification protocol was as follows: initial denaturation at 94 °C for 5 min, followed by 30–35 cycles of denaturation at 94 °C for 1 min, annealing at 52–58 °C for 1 min, extension at 72 °C for 1 min, and a final extension at 72 °C for 10 min (Table 8).

### 4.3. Data Analysis

The company-provided sequencing data were imported into SeqMan for peak comparison, sequence assembly, and correction. Subsequently, after assembling the sequences of the three fragments, they were imported into PhyloSuite version 1.2.3 for sequence alignment and correction [36]. DnaSP version 6.1 was used to calculate conventional indices for *P. discoidea* [37], such as haplotype diversity (*H_d_*), nucleotide diversity (*P_i_*), gene flow (*N_m_*), and differentiation coefficients (*N_st_* and *G_st_*), among others [38]. The size of *N_st_* and *G_st_* was used to determine whether there was a genealogical geographic structure among populations. When *N_st_* was greater than *G_st_* and *P* was less than 0.05, haplotypes with similar phylogenetic relationships were distributed within the same population, indicating that there was an obvious lineage structure among the populations. When *N_st_* was equal to *G_st_*, the phylogenetic relationships among haplotypes across populations were similar. When *N_st_* was less than *G_st_*, it indicated that haplotypes with similar phylogenetic relationships existed in different populations, and there was no lineage structure [39]. Arlequin version 3.5 was used for molecular ANOVA, and 1000 non-parametric permutations were used for significance testing to estimate the distribution of variance within and among populations and the genetic differentiation index (*F_st_*) among populations of *P. discoidea* and further infer the main factors influencing the genetic differentiation in *P. discoidea* [40]. The value of *F_st_* ranges from 0 to 1. When the *F_st_* is between 0 and 0.05, it indicates that genetic differentiation is low, when the *F_st_* is between 0.05 and 0.25, it indicates a moderate degree of genetic differentiation, and when the *F_st_* is greater than 0.25, it represents a significant level of genetic differentiation [41].

PopArt version 1.7 was used to construct the haplotype network diagram among populations to explore the relationships among various haplotypes. Additionally, the geographic distribution of haplotypes was mapped using ArcGIS version 10.8 [42]. PhyloSuite version 1.2.3 was used to construct haplotype phylogenetic trees of *P. discoidea* using the maximum likelihood and Bayesian methods. For the *trnD–E*, *rps16*, and *rpoB* sequences, the best nucleotide substitution model for the maximum likelihood method was GTR2 + ML + R2, while the best nucleotide substitution model for the Bayesian method was HKY + F + G4. For the ITS sequence, the nucleotide substitution model for the maximum likelihood method was TIM-R3, and for the Bayesian method, it was JC-I-G.

BEAST version 1.8.4 [43] was used to estimate the divergence times of *P. discoidea* by using the two-step method combining horizontal phylogenetic trees and haplotype trees, combined with the calibration points obtained from fossils, literature, and tree-building calculations. The molecular clock model chosen was the uncorrelated lognormal relaxed clock model. The results were visualized and edited using ITOL version 5 and FigTree version 1.4.4 [44,45,46]. DnaSP version 6 was used to conduct pairwise mismatch distribution analysis and neutrality tests on *P. discoidea* to examine whether it had experienced population expansion or bottleneck effects [47]. If Fu’s *F_s_* value and Tajima’s D value in the neutrality tests were both negative, it indicated that the *P. discoidea* population had recently undergone a rapid expansion event. Conversely, positive values suggested that the population had experienced a contraction event or bottleneck effect. Mismatch analysis focused on the distribution of nucleotide differences among different haplotypes. By observing the fit between the expected value curve and the observed value curve, as well as whether the overall curve was unimodal or multimodal, historical events recently experienced by the population were inferred [16].

## 5. Conclusions

This study evaluated the population variation, genetic diversity, phylogenetic structure, and dynamic history of *P. discoidea* by using three matrilineal inherited cpDNA fragments and biparentally inherited nuclear ITS sequences. We found high genetic diversity and the existence of phylogenetic structure in *P. discoidea*. One lineage was the central region of Anhui and the western region of Hubei. The other lineage was the Jiangsu region and the Zhejiang region. This study provides insights into the population variation, genetic diversity, phylogenetic structure, and dynamic history of *P. discoidea*. The findings offer a theoretical foundation for the protection and utilization of germplasm resources of *P. discoidea*. Due to the ambiguous information of some samples, accurate geographic information could not be obtained, resulting in a relatively small overall sample from Jiangxi and Anhui, so population genetic variation and differentiation were not fully verified. With the decrease in sequencing costs and the continuous advancement in sequencing techniques, we expect to use whole-genome resequencing and other methods to conduct in-depth studies on the population variation and historical dynamics of *P. discoidea*. This will be aimed at providing more accurate data support for the conservation and utilization of *P. discoidea* germplasm resources.

## Figures and Tables

**Figure 1 plants-13-02535-f001:**
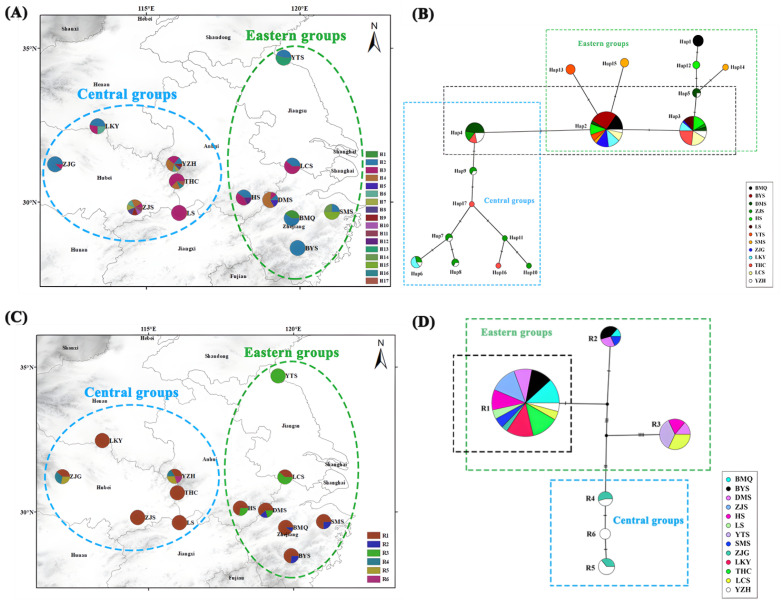
(**A**) Geographical distribution map of the haplotypes of *P. discoidea* based on cpDNA sequences. (**B**) Network diagram of the TCS haplotypes of *P. discoidea* based on cpDNA sequences. (**C**) Geographical distribution map of the ribotypes of *P. discoidea* based on nrDNA sequences. (**D**) Network diagram of TCS ribotypes of *P. discoidea* based on nrDNA sequences.

**Figure 2 plants-13-02535-f002:**
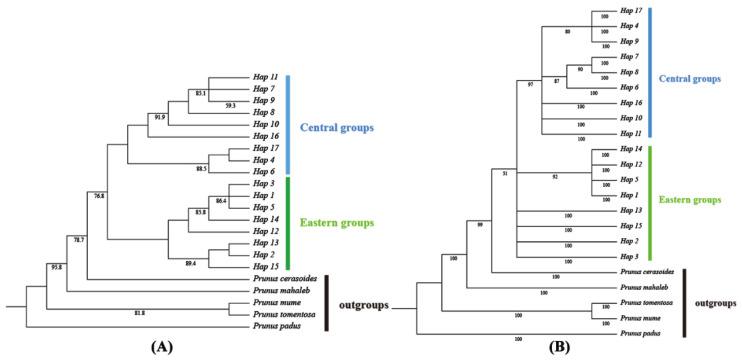
(**A**) ML tree of *P. discoidea* haplotypes based on cpDNA sequences. (**B**) BI tree of *P. discoidea* haplotypes based on cpDNA sequences.

**Figure 3 plants-13-02535-f003:**
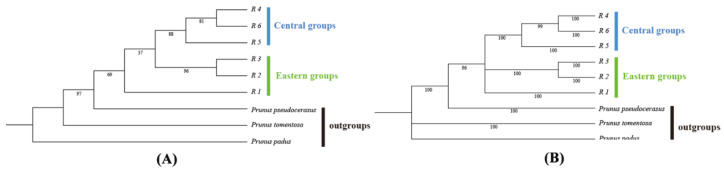
(**A**) ML tree of *P. discoidea* ribotypes based on nrDNA sequences. (**B**) BI tree of *P. discoidea* ribotypes based on nrDNA sequences.

**Figure 4 plants-13-02535-f004:**
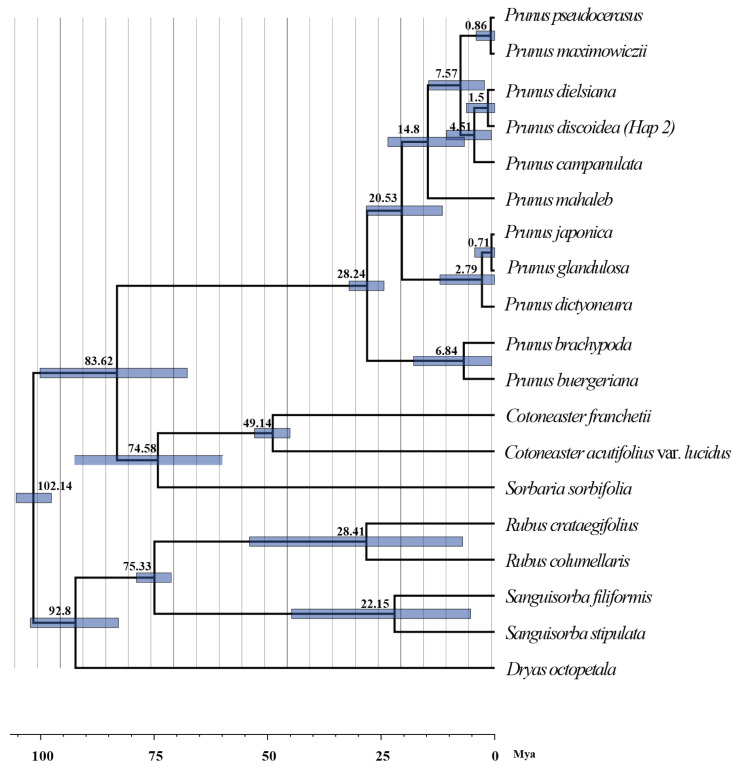
Phylogenetic tree of Rosaceae based on chloroplast DNA (*rps*16) and four fossil dates.

**Figure 5 plants-13-02535-f005:**
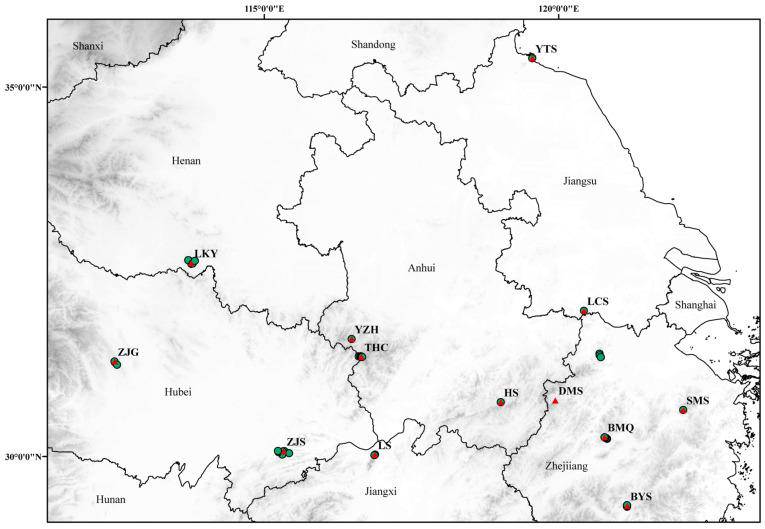
Sampling points and population distribution points of *P. discoidea*. Note: Green represents the sampling points, and red represents the population distribution points.

**Table 1 plants-13-02535-t001:** Haplotype variation sites of P. discoidea.

Haplotype	Base Mutation Loci								
	*trnD–E*			*rpoB*			*rps16*		
	240	416	509	811	844	1018	1569	1743	1793
Hap1	T	A	G	A	T	C	G	A	C
Hap2	C	.	A	.	C	.	.	G	.
Hap3	C	G	.	.	C	T	.	G	.
Hap4	C	.	A	.	C	T	.	G	.
Hap5	C	G	.	.	C	.	.	G	.
Hap6	.	.	.	.	C	T	.	G	.
Hap7	.	.	.	.	C	T	.	G	A
Hap8	.	.	.	.	C	.	.	G	A
Hap9	C	.	A	.	C	T	.	G	A
Hap10	.	G	.	.	C	.	.	G	A
Hap11	C	G	.	.	C	T	.	G	A
Hap12	C	.	.	.	C	.	.	G	.
Hap13	C	.	A	.	C	.	T	G	.
Hap14	C	.	.	.	C	.	.	.	.
Hap15	C	.	A	G	C	.	.	G	.
Hap16	.	.	.	.	C	.	.	.	.
Hap17	C	.	.	.	C	T	.	G	.

Note: Periods represent the base of the Hap1 site.

**Table 2 plants-13-02535-t002:** Genetic diversity index and geographical information of *P. discoidea* based on cpDNA sequences.

	Population Code	*H_d_*	*P_i_*	Sample Size	Haplotype Distribution
1	BMQ	0.498	0.00102	32	H1(13)H2(19)
2	BYS	0.000	0.00000	34	H2(34)
3	DMS	0.667	0.00066	34	H2(4)H3(4)H4(20)H5(6)
4	HS	0.652	0.00077	33	H2(13)H3(14)H12(6)
5	YTS	0.523	0.00027	18	H2(8)H13(10)
6	SMS	0.663	0.00069	16	H2(4)H14(4)H15(8)
7	LCS	0.492	0.00102	26	H2(10)H3(16)
8	ZJS	0.844	0.00113	32	H3(3)H4(7)H6(4)H7(4) H8(4)H9(4)H10(3)H11(3)
9	LS	0.000	0.00000	11	H3(11)
10	ZJG	0.209	0.00032	18	H2(16)H3(2)
11	LKY	0.659	0.00094	32	H2(15)H3(9)H6(8)
12	THC	0.575	0.00068	32	H3(20)H4(6)H16(3)H17(3)
13	YZH	0.891	0.00104	30	H2(4)H3(7)H4(9)H5(2) H6(2)H7(2)H8(2)H9(3)
Eastern		0.703	0.00087	193	H1(13)H2(92)H3(34) H4(20)H5(6) H12(6)H13(10)H14(4) H15(8)
Central		0.807	0.00103	155	H2(25)H3(48)H4(22) H5(2)H6(14)H7(6) H8(6)H9(6)H10(3)H11(3)H16(3)H17(3)
Mean		0.546	0.000696		
Total		0.782	0.00104	348	

**Table 3 plants-13-02535-t003:** Ribotypes and variation sites of *P. discoidea*.

Ribosome	Base Mutation Loci												
	84	132	133	135	137	153	186	221	231	465	485	495	620
R1	G	G	A	G	G	C	C	T	A	C	T	A	T
R2	.	T	T	.	.	.	.	.	.	.	.	.	.
R3	.	.	T	.	A	.	.	C	C	A	G	G	A
R4	.	.	T	T	.	A	T	.	C	.	G	.	A
R5	.	.	T	T	.	.	.	.	C	.	G	.	A
R6	C	.	T	T	.	.	T	.	C	.	G	.	A

Note: Periods represent the base of the Hap1 site.

**Table 4 plants-13-02535-t004:** Genetic diversity index and geographical information of *P. discoidea* based on nrDNA sequences.

	Population Code	*R_d_*	*P_i_*	Sample Size	Ribotype Distribution
1	BMQ	0.175	0.00051	32	R1(29)R2(3)
2	BYS	0.401	0.00116	34	R1(25)R2(9)
3	LCS	0.492	0.00571	26	R1(10)R3(15)
4	DMS	0.561	0.00458	34	R1(21)R2(6)R3(7)
5	HS	0.409	0.00474	33	R1(24)R3(9)
6	YTS	0.000	0.00000	18	R3(18)
7	SMS	0.400	0.00116	16	R1(12)R2(4)
8	LS	0.000	0.00000	11	R1(11)
9	ZJG	0.699	0.00560	18	R1(6)R4(6)R5(5)
10	LKY	0.000	0.00000	32	R1(32)
11	ZJS	0.000	0.00000	32	R1(32)
12	THC	0.000	0.00000	32	R1(32)
13	YZH	0.756	0.00537	30	R1(10)R4(5)R5(9)R6(6)
Eastern		0.530	0.00489	193	R1(122)R2(22)R3(50)
Central		0.348	0.00336	155	R1(123)R4(11)R5(14)R6(6)
Mean		0.318	0.00247		
Total		0.478	0.00451	348	

**Table 5 plants-13-02535-t005:** Analyses of molecular variance (AMOVAs) based on cpDNA and nrDNA data for populations of *P. discoidea*.

Source of Variation	d.f.	Sum of Squares	Variant Components	Percentage of Variation	Fixation Index
	Chloroplast DNA Fragments				
All groups					
Among populations	12	120.807	0.35323	34.26	*F_st_ *= 0.34264
Within populations	335	227.026	0.67769	65.74	
Central					
Among populations	5	36.409	0.25649	24.45	*F_st_ *= 0.24453
Within populations	149	118.069	0.79241	75.55	
Eastern					
Among populations	6	52.182	0.29754	33.68	*F_st_ *= 0.33684
Within populations	186	108.957	0.58579	66.32	
Eastern and Central					
Among groups	1	32.216	013640	12.47	*F_sc_ *= 0.29216 *F_st_ *= 0.38043 *F_ct_ *= 0.12470
Among populations within groups	11	88.591	0.27972	25.57	
Within populations	335	227.026	0.67769	61.96	
	nuclear DNA fragments				
All groups					
Among populations	12	274.754	0.83160	57.65	*F_st_ *= 0.57621
Within populations	335	264.838	0.70560	42.35	
Central					
Among populations	5	92.266	0.70636	54.83	*F_st_ *= 0.54835
Within populations	149	86.689	0.58180	45.17	
Eastern					
Among populations	6	145.809	0.85631	47.20	*F_st_ *= 0.47203
Within populations	186	178.150	0.95779	52.80	
Central and Eastern					
Among groups	1	36.679	0.07575	4.57	*F_sc_ *= 0.57791 *F_st_ *= 0.52292 *F_ct_ *= 0.04571
Among populations within groups	11	238.057	0.79078	47.72	
Within populations	335	264.838	0.79056	47.71	

Note: *F_ct_*: Proportion of genetic variation among groups. *F_sc_:* Proportion of genetic variation between populations within groups. *F_st_*: Proportion of genetic variation between populations and groups overall.

**Table 6 plants-13-02535-t006:** Information on *P. discoidea* sampling points.

Province	City	Longitude	Latitude	Province	City	Longitude	Latitude
Jiangxi	Jiujiang	116.0603	29.64585	Hubei	Xianning	114.6116	29.82106
		116.0573	29.64455			114.6262	29.82052
		116.0506	29.63852			114.6975	29.78564
		116.0495	29.63825			114.5980	29.81822
		116.0482	29.63799			114.5883	29.77600
Zhejiang	Lishui	119.9113	28.50320			114.5869	29.77621
	Hangzhou	119.9107	28.49354			114.5301	29.81668
		119.9090	28.49267			114.5265	29.82496
		119.9095	28.49301			114.5234	29.83107
		119.9111	28.49441		Jingmen	112.0240	31.21449
		119.9109	28.49600			112.0178	31.22911
		119.9069	28.49807			112.0562	31.18117
		119.9122	28.50784		Huanggang	115.9719	30.99700
		119.9130	28.51143			115.9970	30.99309
		119.9135	28.51229			116.0092	30.99434
		119.9114	28.51227			116.0107	30.98784
		119.9111	28.51197			116.0250	30.98490
		119.9109	28.51256			116.0226	30.98254
		119.9122	28.51424			116.0258	30.98203
		119.7562	29.44729	Henan	Nanyang	113.3932	32.46884
		119.7467	29.45877			113.3566	32.51996
		119.7271	29.46597			113.3492	32.52327
		119.7225	29.47247			113.4165	32.47025
		119.7213	29.47398			113.4575	32.50649
		119.7189	29.47520	Jiangsu	Lianyungang	119.4586	34.70469
		119.7194	29.47445			119.4578	34.70529
	Yuyao	121.0254	29.67678			119.4580	34.70554
	Huzhou	119.8398	30.61118		Yixing	119.6931	31.21817
		119.8372	30.58625				
		119.8513	30.55984				
		119.8506	30.55914				
Anhui	Huangshan	118.1552	30.14521				
	Lu’an	115.8897	31.24339				

**Table 7 plants-13-02535-t007:** Geographical information on *P. discoidea* populations.

Sample Code	Sampling Site	Sample Size	Longitude and Latitude	Altitude/m
DMS	Daming Mountain, Lin’an District, Zhejiang Province	34	119.02536 E 30.06924 N	800
HS	Huang Mountain, Huangshan City, Anhui Province	33	118.15517 E 30.14521 N	850
BYS	Baiyun Mountain, Lishui City, Zhejiang Province	34	119.91132 E 28.50266 N	333
SMS	Siming Mountain, Yuyao City, Zhejiang Province	16	121.02541 E 29.67678 N	847
BMQ	Eightmu Hill, Jiande County, Zhejiang Province	32	119.71899 E 29.4761 N	351
LCS	Longchi Mountain, Yixing City, Jiangsu Province	26	119.69308 E 31.21817 N	324
YTS	Yuntai Mountain, Lianyungang City, Jiangsu Province	18	119.45889 E 34.70508 N	157
YZH	Yanzi River, Lu’an City, Anhui Province	30	115.88972 E 31.24339 N	568
THC	Taohua Chong, Huanggang City, Hubei Province	32	116.01271 E 30.98719 N	522
ZJS	Zengjia Mountain, Xianning City, Hubei Province	32	114.61163 E 29.82106 N	118
ZJG	Zhangjia Gully, Jingmen City, Hubei Province	18	112.01784 E 31.22911 N	285
LS	Lu Mountain, Jiujiang City, Jiangxi Province	11	116.05310 E 29.64308 N	222
LKY	Tongbo County, Nanyang City, Henan Province	32	113.39161 E 32.48135 N	169
total		348		

**Table 8 plants-13-02535-t008:** Primer sequence and annealing temperature.

Primer Name	Primer Sequence	Tm/°C	Number of Cycles
*rps16*	F: GTGGTAGAAAGCAACGTGCGACTT R: TCGGGATCGAACATCAATTGCAAC	52	30
*rpoB*	F: AAGTGCATTGTTGGAACTGG R: CCCAGCATCACAATTCC	56	35
*trnD–E*	F: ACCAATTGAACTACAATCCC R: AGGACATCTTCAAGGAG	56	35
ITS	F: TCCTCCGCTTATTGATATGC R: GGAAGGAGAAGTCGTAACAAGG	58	35

## Data Availability

The data presented in this study are available upon request from the corresponding author due to privacy restrictions.

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
