# Peer review of "Phylogeography and Population Variation in *Prunus discoidea* (*Prunus* subg. *Cerasus*) in China"

_plants, 2024, doi:10.3390/plants13172535_

Round 1

Reviewer 1 Report

Comments and Suggestions for Authors

The manuscript reports a phylogeographic study on the native cherry blossom species Prunus discoidea in China. The authors used chloroplast DNA (cpDNA) and ribosomal DNA (rDNA) markers to analyze the genetic variation and evolutionary history of this species. The findings have implications for the conservation and sustainable management of this germplasm resource. However, there are several aspects of the manuscript that could be improved. Below are my comments and suggestions: 
Main Comments:
1. The authors should provide clear definitions of the genetic diversity indices, such as Hd and Rd. Line 215: The genetic diversity (Ht) or (Hd)?

2. The differentiation index (Fst) is presented, but the interpretation of these values could be expanded upon. Specifically, the biological significance of the observed differentiation levels and how they relate to the geographic structure should be discussed.

3. The mismatch distribution analysis suggests a recent population expansion, but this conclusion is not supported by the neutrality tests. The authors need to reconcile these contradictory findings and provide a more robust discussion of the demographic history of P. discoidea.
4. The authors should consider discussing the potential factors driving the observed genetic structure, such as historical climate changes, ecological factors, and human activities.
5. The authors should provide a more detailed explanation of the methods used, including the sampling strategy and the choice of molecular markers. Provide more details about the sample collection sites and the number of individuals sampled per population.
6. Consider adding a map showing the distribution of the sampled populations to help visualize the geographic structure.
7. The authors should consider discussing the limitations of their study, such as the potential for sampling bias or the representation of the entire species range.

Author Response

Manuscript. Number.: plants-3176728

Title: Phylogeography and Population Variation of Prunus discoidea (Prunus subgen. Cerasus) in China

Thank you for giving us the opportunity of reviewing our manuscript. Those comments are all valuable and very helpful for revising and improving our paper. Our replies to the reviewers comments are reported hereafter. All changes with respect to the previous submission have been highlighted in the Word file of the manuscript.

Comments 1The authors should provide clear definitions of the genetic diversity indices, such as Hd and Rd. Line 215: The genetic diversity (Ht) or (Hd)?

Response 1: Thank you for pointing this out. Haplotype diversity(Hd) referred to the variety and frequency of distribution of haplotypes within a sample, measuring the richness of haplotypes in a population and the evenness of their distribution. Ribosomal diversity (Rd) referred to the genetic diversity of ribosomal genes across different individuals or populations. Both help to reveal the structure of the population and the genetic variation.

The entire text has been consistently using Hd

Comments 2The differentiation index (Fst) is presented, but the interpretation of these values could be expanded upon. Specifically, the biological significance of the observed differentiation levels and how they relate to the geographic structure should be discussed.

Response 2: Thank you for pointing this out. We agree with this comment. Therefore, I made the following changes in the Data analysis: The size of Nst and Gst was used to determine whether there was genealogical geographic structure between populations. When Nst was greater than Gst and P was less than 0.05, haplotypes with similar phylogenetic relationships were distributed within the same population, indicating that there was an obvious lineage structure among the populations. When Nst was equal to Gst, the phylogenetic relationships among haplotypes across populations were similar. When Nst was less than Gst, it indicated that haplotypes with similar phylogenetic relationships existed in different populations, and there was no lineage structure.

This change can be found on page 14, paragraph 2,line 358-366.

The value of Fst ranges from 0 to 1. When the Fst is between 0 and 0.05, it indicates that genetic differentiation is low; when the Fst is between 0.05 and 0.25, it indicates a moderate degree of genetic differentiation; and when the Fst is greater than 0.25, it represents a significant level of genetic differentiation.

This change can be found on page 14, paragraph 2,line 370-373.

Comments 3The mismatch distribution analysis suggests a recent population expansion, but this conclusion is not supported by the neutrality tests. The authors need to reconcile these contradictory findings and provide a more robust discussion of the demographic history of P. discoidea.

Response 3: Thank you for pointing this out. We agree with this comment. Therefore, I made the following changes in the Historical dynamics of P. discoidea group: Based on neutrality tests for P. discoidea populations and geographic groups, the results indicated that neither of the two molecular markers detected population expansion or contraction events. However, the mismatch distribution analysis based on cpDNA molecular markers for P. discoidea population and geographic groups showed an unimodal curve. Both the SSD value of 0.01938 (P-value=0.18000>0.05) and the Hrag value of 0.05561 (P-value=0.33000>0.05), are consistent with the hypothesis of the population expansion model. This suggested that P. discoidea population had experienced a population expansion event. The results contradicted those of the neutrality tests. However, according to Table S1, the population size of P. discoidea was θ0=0.00176 before the outbreak, and θ1=5.33203 after the outbreak. The change in effective population size (θ01=5.33203-0.00176) was large, so it was considered that P. discoidea had recently experienced a population expansion event. This result was consistent with the phylogeographic study of Xanthopappus subacaulis in the northeastern Qinghai-Tibet Plateau conducted by Zhang Yang et al.

This change can be found on page 10, paragraph 4,line 280-293.

Comments 4The authors should consider discussing the potential factors driving the observed genetic structure, such as historical climate changes, ecological factors, and human activities.

Response 4: Thank you for pointing this out. We agree with this comment. Therefore, I made the following changes in the Genetic diversity and population genetic structure: However, both markers indicated that P. discoidea exhibited a high level of genetic diversity, which is presumed to be related to the growth environment and the distribution of its habitats. P. discoidea was distributed in Central and Eastern China, located on the third step of China’s geographical terrain, and was concentrated in the middle and lower reaches of the Yangtze River. The region was characterized by flat terrain with no mountainous barriers. At the same time, the warm and humid climate conditions maintained their genetic diversity, and the activities of birds and humans made hybridization and self-pollination within or between neighboring populations possible.

This change can be found on page 9, paragraph 1,line 242-249.

Comments 5The authors should provide a more detailed explanation of the methods used, including the sampling strategy and the choice of molecular markers. Provide more details about the sample collection sites and the number of individuals sampled per population.

Response 5: Thank you for pointing this out. We agree with this comment. Therefore, I made the following changes in the Plant materials: The distribution data of P. discoidea is primarily based on the Chinese Virtual Herbarium(CVH:https://www.cvh.ac.cn/) and published academic papers. For distribution points with accurate specimen records but lacking latitude and longitude data, LocaSpaceViewer (http://www.locaspace.cn/) was used to ascertain the coordinates, thereby enhancing the precision of the specimen information. DIVA-GIS was used to filter the obtained data, deleting duplicate records and those with collection points that were too close to each other. From 2020 to 2022, a total of 348 samples from 13 populations of P. discoidea were collected through two consecutive years of field investigation and sample collection (Table 6, Table 7, Figure 5).

This change can be found on page 11, paragraph 2, line 308-316, 320, 323-324.

Comments 6Consider adding a map showing the distribution of the sampled populations to help visualize the geographic structure.

Response 6: Thank you for pointing this out. We agree with this comment. Therefore, I made the following changes in the Plant materials. This change can be found on page 16, line 323-324.

Figure 5. Sampling points and population distribution points of P. discoidea. Green represents the sampling points, and red represents the population distribution points.

Comments 7The authors should consider discussing the limitations of their study, such as the potential for sampling bias or the representation of the entire species range.

Response 7: Thank you for pointing this out. We agree with this comment. Therefore, I made the following changes in the Conclusions: Due to the ambiguous information of some samples, accurate geographic information could not be obtained, resulting in a relatively small overall sample size from Jiangxi and Anhui, so population genetic variation and differentiation were not fully verified. With the decrease in sequencing costs and the continuous advancement of sequencing techniques, we expect to use whole-genome resequencing and other methods to conduct in-depth studies on the population variation and historical dynamics of P. discoidea. This will be aimed at providing more accurate data support for the conservation and utilization of P. discoidea germplasm resources.

This change can be found on page 15, paragraph 2, line 407-414.

Reviewer 2 Report

Comments and Suggestions for Authors

Dear Authors,

Please find my comments in the attached file.

Regards

Author Response

Manuscript. Number.: plants-3176728

Title: Phylogeography and Population Variation of Prunus discoidea (Prunus subgen. Cerasus) in China

Thank you for giving us the opportunity of reviewing our manuscript. Those comments are all valuable and very helpful for revising and improving our paper. Our replies to the reviewers comments are reported hereafter. All changes with respect to the previous submission have been highlighted in the Word file of the manuscript.

Comments 1:Do you analyze all genetic structure of the population?

Response 1: Thank you for pointing this out. We agree with this comment. Therefore, we changed the title to Phylogeography and Population Variation of Prunus discoidea (Prunus subgen. Cerasus) in China.

This change can be found on page 1, line 1-2.

Comments 2:Research problem, why you do your work?

Response 2: Thank you for pointing this out. We agree with this comment. Therefore, I made the following changes in the abstract: This study employed phylogeographic analysis to reveal the evolutionary history of P. discoidea to better understand its genetic diversity and structure. This study will provide more accurate molecular insights for the effective conservation and utilization of this germplasm resource.

This change can be found on page 1, line 11-14.

Comments 3:What is your research problem?

Response 3: Thank you for pointing this out. We agree with this comment. Therefore, I made the following changes in the Introduction: Prunus discoidea is a member of the Prunus subg.Cerasus in the Rosaceae family. It is an excellent germplasm resource endemic to China. The branches of P. discoidea are graceful and spreading, with pink flowers that bloom early in the season.

This change can be found on page 2, line 31-33.

Comments 4: Belongs to methods

Response 4: Thank you for pointing this out. We agree with this comment. Therefore, I made the following changes in the Introduction: In this study, based on a comprehensive survey of wild populations of P. discoidea and systematic sampling, this study conducted a phylogeographic analysis of P. discoidea. Chloroplast DNA sequences and nuclear ribosomal internal transcribed spacer (ITS) sequences were used to analyze the genetic diversity and genetic structure of P. discoidea and employed an integrative method to trace its evolutionary history. These findings will provide a theoretical foundation for future strategies related to the conservation and utilization of P. discoidea resources.

This change can be found on page 2, line 69-75.

Comments 5:There is no need to do repetition of your results. You can simply refer to the figure or table needed.

Response 5: Thank you for pointing this out. We agree with this comment. Therefore, I made the following changes in the Discussion: As shown in Table 5, based on three cpDNA fragments, the genetic variation among populations was lower than that within populations. Based on the ITS fragment, the genetic variation among populations was higher than that within populations. The genetic differentiation coefficients for both molecular markers reached significant levels, and gene flow among populations of P. discoidea was relatively weak. Therefore, this study suggested that the variation in the population of P. discoidea mainly arose from the variation within populations.

This change can be found on page 10, paragraph 2,line 250-254.

Comments 6:Repetition of yor results. Please compare your findings with other authors' findings.

Response 6: Thank you for pointing this out. We agree with this comment. Therefore, I made the following changes in the Discussion: Based on three cpDNA fragments, the results for P. discoidea populations and two geographic groups separately showed that the genetic differentiation coefficients reached significant levels. This finding was consistent with the genetic differentiation parameters of nrDNA markers, indicating the presence of phylogeographic structure within the geographic groups and populations of P. discoidea. The research results were similar to those of previous studies on P. serrulate, P. dielsiana, and P. conradinae. This result was consistent with the habits of P. discoidea, which in its natural state tended to individual scattered distribution.

This change can be found on page 10, paragraph 2,line 258-263.

Comments 7: Please compare your findings with other authors' findings.

Response 7: Thank you for pointing this out. We agree with this comment. Therefore, I made the following changes in the Discussion: This formed two distinct lineages in the eastern and central regions. The research results were consistent with those of previous studies on P. serrulata.

This change can be found on page 10, paragraph 3,line 277-278.

Comments 8:What about the replication of your results? I mean that you have to have repeated measurements to do crosschecking between them.

Response 8: Thank you for pointing this out. We agree with this comment. Therefore, I made the following changes in the Plant materials: Within each population, 10 to 35 individuals were randomly selected, each at least 30 meters apart. For each individual, 5 to 10 mature, healthy, and intact small leaves were collected. Then, the samples were rapidly placed in silica gel for drying.

This change can be found on page 11, paragraph 2,line 316-319.

Comments 9:But did you do it?

Response 9: Thank you for pointing this out. We agree with this comment.Yes,I did. The samples were placed in a -20°C freezer to prevent degradation of genetic material, thereby effectively maintaining the integrity of genetic substances. Therefore, I made the following changes in the Plant materials: Finally, the samples were put into the refrigerator at -20℃ for use.

This change can be found on page 11, paragraph 2,line 319.

Comments 10:What data was provided by company? You had your own experiment and dit extraction or not?

Response 10: Thank you for pointing this out. In this experiment, we employed both a polysaccharide-polyphenol kit and a modified CTAB method to extract DNA from P. discoidea. Following extraction, DNA quality was assessed using 1% agarose gel electrophoresis, and samples that did not meet the quality criteria were excluded. We sent the samples that passed quality checks to the company for sequencing. The company provided the raw data, which we then filtered, assembled, and annotated.

Comments 11:This does not come from the results, this is rather discussive sentence. So please discuss it in discussion part

Response 11: Thank you for pointing this out. We agree with this comment. Therefore, I made the following changes in the Conclusions: We found high genetic diversity and the existence of phylogenetic structure in P. discoidea. One lineage was the central region of Anhui and the western region of Hubei. The other lineage was Jiangsu region and the Zhejiang region. This study provided insights into the population variation, genetic diversity, phylogenetic structure, and dynamic history of P. discoidea.

This change can be found on page 15, paragraph 2,line 401-404.
